# Medical Misadventures as Errors and Mistakes and Motor Vehicular Accidents in the Disproportionate Burden of Childhood Mortality among Blacks/African Americans in the United States: CDC Dataset, 1968–2015

**DOI:** 10.3390/healthcare12040477

**Published:** 2024-02-15

**Authors:** Laurens Holmes, Michael Enwere, Robert Mason, Mackenzie S. Holmes, Pascal Ngalim, Kume Nsongka, Kerti Deepika, Gbadebo Ogungbade, Maura Poleon, David T. Mage

**Affiliations:** 1Public Health & Allied Health Sciences Department, DSU-WC, Delaware State University, Dover, DE 19901, USA; 2Lawhols International Scientific Research Consulting, LIS-RC, Bear, DE 19701, USA; mikky89@gmail.com (M.E.);; 3Global Health Equity Foundation, Bear, DE 19701, USA; 4Texas A & M University, College Station, TX 77843, USA; 5Global Health Service Initiatives, Arlington, TX 76014, USA; 6Miami Baptist Hospital, Miami, FL 33176, USA; maurapoleon@gmail.com; 7USEPA, Newark, DE 19716, USA

**Keywords:** infant mortality, racial disparities, medical misadventures, motor vehicle-related mortality, U.S. mortality trends

## Abstract

Purpose: Racial disparities in infant mortality in the United States persist after adjustment for known confounders of race and mortality association, as well as heterogeneity assessment. Epidemiologic and clinical data continue to show the survival disadvantages of Black/AA children: when Black/AAs are compared to whites, they are three times as likely to die from all-cause mortality. The persistent inability to remove the variance in race–mortality association is partly due to unobserved, unmeasured, and residual confounding, as well as implicit biases in public health and clinical medicine in health equity transformation. This current epidemiologic-perspective explanatory model study aimed to examine the possible explanation of racial differences in U.S. infant mortality using medical misadventures as errors and mistakes, and infants’ involvement in motor vehicular traffic accidents. Materials and Method: Using CDC WONDER ecologic data from 1968 to 2015, we assessed the infant mortality-rate ratio and percent change associated with medical misadventures as well as motor vehicular accidents or trauma. The rate ratio and percent change were estimated using stratification analysis and trend homogeneity, respectively. Results: There was a Black–white racial difference in medical misadventures during the study period. Relative to the years 1968–1978 (rate ratio [RR], 1.43), there was a steady increase in the mortality-rate ratio in 1979–1998 (52%, RR = 1.52), and mortality was more than two times as likely in 1999–2015 (RR = 2.37). However, with respect to motor vehicular accident/trauma mortality, the mortality ratio, although lower among Blacks in 1968–1978 (RR, 0.92), increased in 1979–1998 by 27% (RR = 1.27) but decreased in 1999–2015 (RR, 1.17), though there was still an excess of 17% mortality among Black/AAs. The percent change for medical misadventures indicated an increasing trend from 9.3% in 1998 to 52% in 2015. However, motor vehicular-related mortality indicated a positive trend in 1998 (38.5%) but a negative trend in 2015 (−8.4%). Conclusions: There were substantial race differentials or variances in infant mortality associated with medical misadventures compared to traffic accidents, and Black/AA children relative to whites experienced a survival disadvantage. These comparative findings are suggestive of medical misadventures and motor vehicular trauma as potential explanations for some of the persistent Black–white disparities in overall infant mortality in the U.S. From these findings, we recommend a national effort to address these issues, thus narrowing the observed disparities in the U.S. infant mortality burden among Black/AAs.

## 1. Introduction

Epidemiological data consistently show a higher total mortality rate for Black or African American (AA) infants than for white infants in the U.S. [1]. Available clinical and population-based data support the contributing effect of small for gestational age (SGA), implying a lower birth weight among Black/AAs relative to whites. SGA is associated with failure to thrive, multiple pathologies, and poor survival rates. The observed excess SGA among Black/AA infants remains an effective measure and modifier of the pathway to equality in infant mortality between races. 

There are several confounding factors in the causal pathway of race and infant mortality that serve as an attempt to explain the causal association between race and infant mortality. SGA may be associated with nicotine use during pregnancy, maternal obesity, late prenatal care (access and care utilization), alcohol use, low maternal education, and disadvantaged neighborhood environmental factors. A recent investigation of gas drilling and proximity to residences in Pennsylvania, USA, observed low birth weights among residents within one kilometer of the draining sites, relative to women residing three kilometers away from gas drilling sites [2]. The disproportionate distribution of the potential causal risk factors for SGA and survival provides some explanation for the excess infant mortality among Blacks/AAs in the U.S., but this cannot be fully explained. 

Previous studies have observed that from 1968 to 2015 in the U.S., this discrepancy existed for all chapters of the WHO International Classifications of Diseases (ICD), except for neoplasms [3]. However, by controlling for differences between non-physiological risk factors, such as maternal age, socio-economic status as income or poverty, and education, as well as physiological factors, such as gestational age and birth weight, there remains an estimated two-fold excess mortality risk for Black/AA infants compared to their white counterparts [4]. The persistent variance or disparities found after controlling for these surrogate causal risk factors, or confounding factors, as distortions in the association between race and infant mortality, is indicative of unmeasured and residual confounding, since no matter how sophisticated the statistical software used for multivariable modeling is, residual confounding persists [5]. 

Since subpopulation differences persist for virtually all causes of infant death, there is a need to examine a common internal risk factor that could play a role in all classes of disease that would be more prevalent in AA than in white cases. The most obvious possibility could arise, perhaps, from the difference in the genes that influence skin reflectance and darkness, such as the melatonin system, or MC1R, which is important for darker skin [6]. However, there is no indication in the literature that this possibility exists. An additional explanation resides in epigenomic alterations and gene expression with respect to the prognostic biomarkers. However, such data are not yet available for the risk-specific characterization of infant mortality. 

Therefore, we examined the ICD cause of death that should theoretically occur at the same rate, independent of the infant race. We chose Misadventures to Patients During Surgical and Medical Care in the U.S., 1968–2015. In effect, the fatal medical errors that occur among healthcare providers, such as the slipping of a scalpel or administration of an incorrect dosage or drug, are independent of the race of the infant patient. Such errors may be explained by the training, skills, and experience of the medical staff treating infants. Prior to this study, an approximately two-fold higher mortality from medical misadventures was observed among Black/AA infants compared to white infants.

With this observation, this study considered other trauma causes of death that should also be independent of the infant’s race. We decided to investigate the ICD subclass of car occupants injured in collision with a car, pick-up truck, or van, and passenger injury in a traffic accident. Additionally, the probability of a vehicle being involved in a potentially fatal-to-non-driver accident is independent of the race of the infant passenger. 

## 2. Materials and Method

### 2.1. Study Design

A retrospective study was used to assess pre-existing aggregate-level data on the association between race and mortality from surgical and medical misadventures as well as passenger mortality in motor vehicular accidents. The ecological design used in this study employed aggregated data instead of individual data, making causal inference challenging because of the absence of individual-level information. This approach involves analyzing the statistical relationships between exposures and outcomes using group-level or aggregated data. Although causality may be difficult to establish at the ecologic level, the direction of observations is confirmed by subsequently incorporating individual data into the process of causal inference. Ecological studies, while informative at the group level, require the integration of individual-level information for a more comprehensive understanding of causal relationships between exposures and outcomes. 

### 2.2. Data Source 

The aggregate data used in this assessment were obtained from the Centers for Disease Control and Prevention (CDC) [7]. The CDC collects data on several health conditions including infant mortality distribution and determinants, but not infant race, which requires an implicit assumption that might not be valid (for infants chosen in this study, the CDC only reported the birth mother’s self-identified race as Black or white. If she reports biracial Black and white parentage, the NCHS “bridges” between them and assigns her to either Black or white via an empirical algorithm. In this study, we used the CDC-reported maternal race as the infant’s race if, and only if, the mother was monoracial and the unmentioned father was the identical race to the mother. For older ages, the CDC reports the subject’s race as recorded by the Funeral Director on the death certificate, which was obtained from the next-of-kin if available, or by personal observation if not, and bridging was applied if they were biracial). Even so, the data source—wonder.cdc.gov—reflects a unique aggregate of racial data for infant mortality, sudden infant death syndrome, and related pathologies, subject to the imprecision of multiracial parentage reported as a single race only. 

The variables utilized in this study include gender, motor vehicular accident, race/ethnicity, medical error and misadventure, timing as calendar years, mortality, and mortality trends. The racial/ethnic variables were classified as Blacks, African Americans (AA), whites, and Hispanics. The calendar year was categorized according to the duration of medical errors, mistakes, and motor vehicular accidents. Medical errors and mistakes were further classified as medical misadventures, indicating instances of patient negligence by healthcare providers (physicians and nurses). This negligence may be explained in part as clinician and implicit bias. 

### 2.3. Data Assessment and Rationale

We reviewed ICD classes from the trauma section to determine whether Black/AAs had a greater risk of mortality from an external cause independent of infant race. The first section was chosen for the subchapter titled Misadventures to Patients during Surgical or Medical Care. Therefore, the mortality rate related to this cause should be approximately the same between AA and white patients if they have similar quality medical care. 

Next, we selected the ICD subclass Car Occupant Fatally Injured in Collisions with Car, Pick-up Truck, or Van for a comparable assessment. We postulated that the probability of an accident, and its severity in traffic, is independent of the race of the passenger, as are now the response time of the emergency vehicle and the medical response at the scene or in the hospital emergency room. (NB: During the Jim Crow era in the Southern United States, Black and white victims of accidents or medical emergencies were often taken to different hospitals, depending on their race. This was because Black ambulances were typically called for Black victims, and white ambulances were called for white victims [8]). 

With this postulate, we expected the mortality rate to be approximately the same if Black/AA and white infants had similar quality emergency medical care and spent approximately the same amount of time in a car as passengers.

### 2.4. Data Analysis

With the aggregate data on the mortality rates of Black/AA and white individuals available for rate- and risk-ratio estimation, we estimated the mortality-rate ratio using: rate in the exposed (R_exposed_)/rate in the unexposed (R_unexposed_). Substituting: I.The infant mortality rate in Black/AA infants compared to the mortality rate in whites associated with medical misadventures;II.The infant mortality rate in Black/AA infants compared to the mortality rate in whites associated with vehicle occupants in traffic accidents.

The percentage change (PC) in the infant mortality-rate ratio experience, comparing the year-group cohort, namely 1968–1978, 1979–1998, and 1999–2015, was estimated by: (current or final period or year)–(initial or previous period or year), divided by (initial period), multiplied by 100. 

Substituting: (1)RR(1979–1998) − RR (1968–1978)/RR(1968–1978) × 100 (% Change, 1979–1998).(2)RR(1999–2015) − RR (1968–1978)/RR(1968–1978) × 100 (% Change, 1999–2015).

### 2.5. Random Error Quantification and Sampling Variability (p-Value)

The ecologic data utilized in this study require the assessment of biologically and clinically meaningful differences in determining racial differentials or disparities in medical errors and mistakes as well as mortality associated with motor vehicular accidents. However, owing to the large sample size used in this assessment, random error quantification, implying sampling variability and the generalizability of these findings, remained reliable (*p* < 0.05, 5%). 

## 3. Results

The first row of Table 1 shows our analyses of the compiled infant mortality data of the Centers for Disease Control and Prevention (CDC) as ICD-8 1968–1978, ICD-9 1979–1998, and ICD-10 1999–2015, from surgical and medical care errors [5]. Of all the other ages for the same cause of death from medical error, all age cohorts tabulated (33 of 36 in Table 1) had an AA/white ratio greater than 1. This was in part accurate and reliable since the Black/AA population may receive poorer medical care than the white population of a similar age, at all ages up to 85 years.

Over the 48-year period covered (1968–2015), the AA/white ratio increased by 66% from 1.428 to 2.367, indicating that white infants were apparently receiving relatively better medical care than Black/AA infants. Subsequently, this study examined the older age groups to observe if this was also indicated for them. As shown in Table 1, in virtually every age group, the white subjects had a lower rate of medical mishaps, except for 3 of the 39 cells shown in green, which indicated that there was a consistency that was not caused by chance (χ^2^ (1) = 27.9, *p* < 0.001). The unweighted average of all 39 cells was 1.938, meaning there was a 94% higher risk of having a fatal medical misadventure for Black than white subjects, similar to the two-fold difference reported.

Consequently, this study examined another ICD subchapter for trauma death that we expected would be independent of the recipient’s race. We chose to inspect the mortality of infants when they were passengers in a motor vehicle that had a collision in traffic, which was assumed to be independent of the race of the passenger. Table 2 illustrates, as does Table 1, the mortality rates and ICD codes covering the same period, from 1968 to 2015, for all age groups. As expected, 19 of the 39 cells were <1 (in green) and 20 were >1; if there was no difference, χ^2^_Yates_ (1) = 0, *p* = 1. The unweighted average ratio of all 39 cells was 1.032, compared with 1.938 for medical errors.

### 3.1. Infant Mortality-Rate Ratio Trends: Percent Change

Although not presented in Table 1, the trends in the percent change in infant mortality-rate ratio for Black/AAs and whites indicated an increasing survival disadvantage for Black/AAs in medical misadventures between 1968 and 2015. Concerning car occupancy/passenger accident mortality among infants, the percentage between 1968–1978 and 1979–1998 was positive, implying a survival disadvantage of Black/AAs; however, the trend in survival was negative from 1979–1998 to 1999–2015. 

### 3.2. Percent Change Estimation for Medical Misadventures

The percent change is illustrated by the estimation 1.561–1.428 = 0.133/1.428 = 0.093 × 100 = 9.3%, implying a change from 1968–1978 to 1979–1998. This estimation demonstrates a percent change, 2.367–1.561 = 0.81/1.561 = 0.52 × 100 = 52.0%, implying there was a change from 1979–1998 to 1999–2015 (Figure 1).

### 3.3. Percent Change Estimation for Car Occupancy/Passenger Accident

The percent change is shown by 1.274–0.920 = 0.354/0.920 = 0.385 × 100 = 38.5%, implying positive change from 1968–1978 to 1979–1998. This estimation demonstrates the percent change, 1.167–1.274 = −0.107/1.274 = −0.084 × 100 = −8.4%, implying there was a negative percentage change from 1979–1998 to 1999–2015 (Figure 2). 

## 4. Discussion

This epidemiologic-perspective study aimed to provide a possible explanation for excess infant mortality among Black/AAs compared to white children. Using the CDC WONDER data, we examined the rate ratio and trends by percent-change estimation for temporal trends in the mortality experiences of the two subpopulations, reflecting racial variance or disparities. We postulated that medical misadventures as errors and mistakes in routine care delivery, compared to no-differential care provision in car accidents with respect to passengers, may, in part, provide an explanation for the persistent two-fold excess of or disparities in Black/AA and white infant mortality in the U.S. 

There are a few relevant findings based on these data. First, medical misadventures associated with infant mortality are higher among Blacks than among whites at all ages. Secondly, except for the initial years of investigation, between 1968 and 1978, infant mortality associated with passengers in traffic accidents was only slightly higher among Black/AAs compared to whites. Thirdly, there was an increasing percent change in medical misadventures for Black/AAs. Fourthly, the car occupancy mortality percentage increased from 1968–1978 to 1979–1998 but decreased from 1999 to 2015. 

The observed variance or excess mortality of Black/AA infants relative to white infants in medical misadventures [4] and car passenger accidents may explain in part the observed persistent and perpetual survival disadvantage of Black/AA infants in the U.S. Epidemiologic data continue to implicate clinician bias as well as implicit or unconscious bias in care provision and have been linked to poorer AA outcomes and survival, based on observation in a clinical setting [9]. The absence of inequity of care provided following traffic collision trauma may reflect in clinician and implicit bias that tends toward the poorer prognosis and subsequent survival disadvantage of Black/AAs in routine non-emergency medical treatment. Based on the medical errors and mistakes examined in this study, the trend with respect to African American (AA) Black children increased with respect to medical errors and mistakes, while there was fluctuation in motor vehicular accidents. In effect, the medical error and mistakes trends increased from the beginning to the end of the data acquisition period. With respect to motor vehicular accidents comparing Black to white children, the trend did not reflect positive variability but fluctuated.

We have shown that medical misadventure is associated with race differences in infant mortality when comparing Black/AAs and whites. Medical mistakes remain one of the leading causes of mortality in the U.S. and have been associated with low SES and racial/ethnic minorities—mainly Hispanics and Black/AAs. Minority women, including Black/AA women, are more likely compared to whites to miss appointments due to healthcare costs [10]. The CDC’s WONDER data clearly indicate increasing trends in infant mortality associated with medical errors and mistakes. Because nonfatal medical mistakes and errors, implying patient safety and nosocomial infections, are not commonly reported but tend to occur more in poor and Black/AA populations relative to wealthy (higher social class) and white populations, it is plausible to suspect that a higher magnitude of variance in infant mortality, when comparing Black/AAs to whites, can be explained by medical misadventures. Without formal hypothesis testing, we postulate that the potentially confounding factors, namely surgical and medical misadventures, as well as vehicular occupancy in traffic accidents, may explain, in part, the disproportionate racial differences in U.S. infant mortality.

These results support our previous observation that racial disparities in infant mortality rates between Black and white races in America, causing Black infants to be twice as likely to die as white infants from virtually all causes, are systemic and cannot be easily eliminated. Overall, the shameful American sins of slavery, segregation, and Jim Crow repression have left the Black/AA community an urban underclass today [3,8]. The resulting relative poverty and lower socio-economic status of the Black/AA community have reduced the availability of quality medical care and placed Black/AAs at the mercy of the cruel high-cost system in American medicine today, where the best doctors treat the richest patients rather than the most difficult cases. Further, 5.7% of physicians in the U.S. are Black/AA.

Additionally, lower SES may delay diagnosis and treatment if parents do not recognize an approaching temperature crisis or when there is an inflection; thus, as temperature could go up or down, ‘why not wait to see if it goes down?’ For example, one of the authors (DM) taught at San José State University in 1965 and was a state employee, and his Kaiser Plan health insurance cost only USD 1 co-pay per visit, so he took his babies to Kaiser at the first symptom. With the current healthcare costs, concerns arise with the delayed navigation of the healthcare system by Black/AA, poor, and under-served populations, resulting in later diagnoses and more difficult treatments.

Whereas the observed explanations combine to predispose Black/AAs and the socio-economically marginalized, poverty implies an unhealthy diet, a lack of exercise, and stress, to reflect on the obvious. The interaction of diet with the gene-environment implies that gene-environment interactions result in gene expression via epigenomic alterations. Gene expression contributes to disease causation, prognosis, and mortality. These explanations are suggestive of epigenomic studies in trauma, assessing the observed lower survival disadvantage of Black/AA children in trauma units and ICUs. 

The advantage of this study is the association of medical errors and mistakes as well as car passenger trauma with race differences in infant mortality, but there are some limitations. First, we used aggregate data, which limits causal inference. Secondly, as we used an aggregate dataset, we were unable to control for potential confounding factors. However, we do not surmise that the implication of higher AA/white mortality from medical errors and mistakes and equivalent AA/white mortality from car passenger accident trauma is driven solely by these unmeasured confounding factors [5]. Finally, we observed that the inclusion of biracial subjects in the AA and white cohorts may have negatively biased the reported differences between races. 

## 5. Conclusions 

In summary, race differentials in infant mortality, with Black/AAs being disproportionately affected, are explained in part by the excess mortality of Black/AAs associated with medical misadventures (errors and mistakes) as well as motor vehicular traffic accident trauma. This study examines and addresses factors associated with these observations, such as access and utilization of good and reliable medical care and healthcare, as well as cultural and linguistic competency in pediatric settings, in addressing implicit and clinician biases within any pediatric healthcare organization, hospital, or clinic in the United States. With these findings, although ecologic, this study is suggestive of the need for accurate and reliable policy development, implementation, and evaluation at local, state, and federal levels in motor vehicular accident regulation as well as healthcare access and utilization, without clinician and implicit biases in the margination of racial disparities in childhood mortality. 

## Figures and Tables

**Figure 1 healthcare-12-00477-f001:**
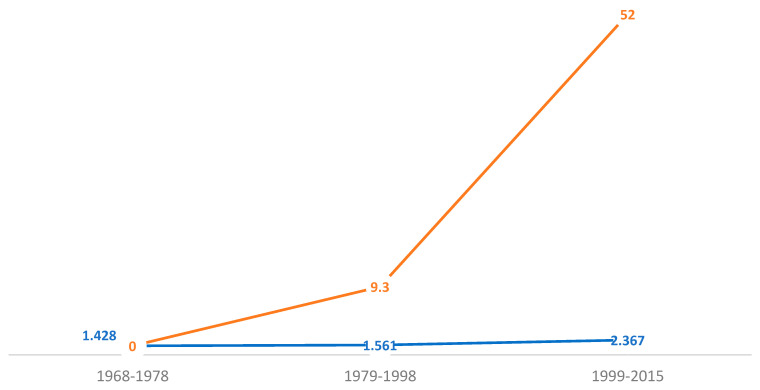
U.S. childhood mortality Black/white prevalence-rate ratio and percent change associated with medical misadventures (errors and mistakes), 1968–2015. Notes and abbreviations: The blue line is indicative of the rate ratio (RR), while the brown line reflects the annual percent change. The medical errors and mistakes prevalence as the cumulative incidence-rate ratio, comparing white and Black/African American children, continues to increase—43.0% (1968–1978), 56.1% (1979–1998)—and is more than two times higher (2.37), 1999–2015. The percent change remains exponential comparing these two sub-populations.

**Figure 2 healthcare-12-00477-f002:**
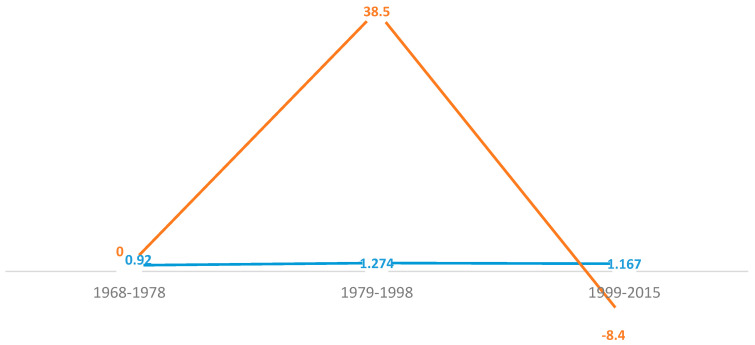
U.S. childhood mortality prevalence-rate ratio and percent change as trends associated with vehicular occupancy in traffic accidents, 1968–2015. Notes and Abbreviations: The blue line is indicative of the rate ratio (RR), while the brown line reflects the motor vehicular accident cumulative percent change. The motor vehicular accident prevalence ratio was slightly lower in the first period (0.92), with Black/AAs being 8% less likely to experience this accident, but this increased in the second period (27% increased risk) and the third period (17% increased risk). The percent change significantly lowered in the third period: 1999–2015.

**Table 1 healthcare-12-00477-t001:** Mortality-Rate Ratios and Trends of Black/AAs to whites in the U.S. by Misadventures (Errors and Mistakes) to Patients during Surgical and Medical Care, 1968–2015.

Age Group	Black/AA-to-White Mortality Ratios		
	1968–1978 ICD-8 E930–E936	1979–1998 ICD-9 E870–E876	1999–2015 ICD-10 Y40–Y84
<1 year	1.428	1.561	2.367
1–4	1.725	1.890	2.392
5–9	1.267	2.179	1.590
10–14	0.946	3.270	1.875
15–19	2.179	2.404	1.015
20–24	2.760	2.806	1.354
25–34	3.346	3.480	1.729
35–44	2.957	3.141	1.972
45– 54	2.105	2.345	2.174
55–64	1.584	2.099	2.079
65–74	1.282	1.545	1.750
75–84	0.931	1.304	1.451
85+	0.745	1.211	1.347

Notes and Abbreviations: Data source—https://www.wonder.cdc.gov (accessed on 15 December 2017) [5]. This table reflects medical errors and mistakes as the third leading cause of mortality, as well as racial differentials in health outcome determinants among children < 1 year (infants) and children 1–19 years. The E930–E936, for example, is indicative of the ICD mortality codes, based on medical misadventures.

**Table 2 healthcare-12-00477-t002:** Mortality-Rate Ratios and Trends of Black/AAs to whites in the U.S. associated with car occupants injured in collision with a car, pick-up truck or van, or passengers injured in traffic accidents.

Age Group	Black/AA-to-White Mortality Ratios		
	1968–1978 ICD-8 812.1, 813.1, 814.1, 815.1, 816.1	1979–1998 ICD-9 811.1, 812.1, 813.1, 814.1, 815.1, 816.1	1999–2015 ICD-10 V43.6
<1 year	0.920	1.274	1.167
1–4	1.039	1.142	1.337
5–9	0.879	0.957	1.073
10–14	0.742	0.766	0.942
15–19	0.602	0.539	0.693
20–24	1.081	0.811	1.001
25–34	1.670	1.105	1.168
35–44	1.887	1.292	1.296
45– 54	1.690	1.347	1.379
55–64	1.418	1.123	1.117
65–74	0.970	0.916	0.831
75–84	0.681	0.778	0.647
85+	0.751	0.625	0.606

Notes and Abbreviations: Data source—https://www.wonder.cdc.gov (accessed on 15 December 2017) [6]. The children’s mortality data reflect infants < 1 year and children 1–19 years in this table. The mortality codes such as 812.1, 813.1, 814.1, 815.1, and 816.1 are indicative of the specifics of the motor vehicular accidents, such as collision with a car, pick-up truck, or van.

## Data Availability

These data are available at the CDC wonder database—http://wonder.cdc.gov (accessed on 5 December 2017).

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
