# Peer review of "Medical Misadventures as Errors and Mistakes and Motor Vehicular Accidents in the Disproportionate Burden of Childhood Mortality among Blacks/African Americans in the United States: CDC Dataset, 1968–2015"

_healthcare, 2024, doi:10.3390/healthcare12040477_

Round 1

Reviewer 1 Report

Comments and Suggestions for Authors

Overall comments: The authors did a great job of developing this manuscript. The MS has a clear message, and statistical analysis makes sense; here are a few comments to improve the quality of the paper.

1. Introduction: The authors partially mixed the last three paragraphs of the introduction with the results section of the paper, it is not usual to report the findings in the introduction, I recommend moving the related section to the results/discussion section and highlighting the study objectives in the last paragraph of the introduction.

2. Method section: Please add a section to discuss the outcome and main variables, such as how the racial categories are defined, how AA has been defined, etc.

3. Results sections: As the author mentioned in the MS, they have yet to report the trends of mortality rates, I recommend adding a line chart to present the trends among different racial groups.

4. Fig 1 and 2 could be presented with much higher quality.

5. The dataset and method did not talk about the “gene environment”; lines 267-271 in the discussion section have opened a new area for discussion without any evidence in the data or analysis, I recommend removing these lines or moderating the tone.

Author Response

  1. Introduction: The authors partially mixed the last three paragraphs of the introduction with the results section of the paper, it is not usual to report the findings in the introduction, I recommend moving the related section to the results/discussion section and highlighting the study objectives in the last paragraph of the introduction.                           Authors Response: 

    Thank you very much for your comments and feedback with respect to the introduction section of this manuscript it is unfortunate this was a misplacement of the results in the introduction section of the manuscript. We have addressed this mistake.

  2.  

     Method section: Please add a section to discuss the outcome and main variables, such as how the racial categories are defined, how AA has been defined, etc.

    Authors Response:      

    Thank you very much for this observation. This has been addressed.

  3. Results sections: As the author mentioned in the MS, they have yet to report the trends of mortality rates, I recommend adding a line chart to present the trends among different racial groups.                                              Authors Response: 

    Thank very much for this observation. We have addressed this.

  4. Fig 1 and 2 could be presented with much higher quality.

    Authors Response

    Thanks for the observation. Unfortunately, Microsoft Excel was used to develop these figures. However, the STATA software was not applicable due to the aggregate data utilized in this study. However, we can increase the contrast to enhance the visibility of the figures. 

  5. The dataset and method did not talk about the “gene environment”; lines 267-271 in the discussion section have opened a new area for discussion without any evidence in the data or analysis, I recommend removing these lines or moderating the tone.

     Authors response:

    Thank you very much for this observation. This observation is within the discussion section; the need for data was not be necessary. Based on the gene and environment interaction, upon which we did not obtain any data from the CDC, this observation as a recommendation remains very adequate and reliable.

Reviewer 2 Report

Comments and Suggestions for Authors

Many thanks to the authors for the manuscript. They address a very relevant topic such as inequalities, based on race in relation to two particular situations,  Errors and Mistakes and Motor  Vehicular Accident, finding that important inequalities persist in the indicators.

Below I write some recommendations to strengthen the presentation of your work:

·        Introduction: Lines 92-104, it seems to me that it does not correspond to the introduction since you are already describing the results of their work, and it can be a bit confusing. I recommend leaving this first section as an introduction only.

·        Methods: In the study design, explain that it is an ecological study

·        Results: report total records analyzed and if data were excluded, or how you deal with missing data.

·        Table 1 and 2: Include p value for Mortality Rate Ratio and the Total and the global calculation of each period

·        Preferably specify as a footnote, the meaning of E930-E936 , E870-E876  y Y40-Y84  (table 1), 812.1, 813.1, 814.1, 815.1, 816.1 , 811.1, 812.1, 813.1, 814.1, 815.1, 816.1 , V43.6  (Table 2)

·        Figure 1 and 2: Review table format and scale in which they are presented. For example, the type of line presented in the graph is continuous and in the legend it is dotted.

·        In Figure 2, it is not very clear which percentage points of change are described with those shown 

You wrote: Notes and Abbreviations: The motor vehicular accidents prevalence ratio was slightly lower in the 282 first period (0.92, with blacks/AA, 8% less likely to experience this accident, but increased in the 283 second period (27% increased risk), and third period (17% increased risk). The percent change significantly lowered in the third period, 1999-2015. But in the figure, % change shows only 3 points  (0, 38.5 y -8.4)

·        Discussion: It is important to clearly discuss the quality of the information analyzed

·        Also discuss the reason for the change in percentage change (negative) of the last traffic accident period

Author Response

1. Introduction: Lines 92-104, it seems to me that it does not correspond to the introduction since you are already describing the results of their work, and it can be a bit confusing. I recommend leaving this first section as an introduction only.                                                                                                           Authors Response: 

Thanks immensely for your comment. It is unfortunate that the section was placed in the introduction rather than the result. We have addressed this.

2. Methods: In the study design, explain that it is an ecological study                   Authors' Response: 

Thanks for this observation. We have addressed, kindly find it in the revised manuscript.

3. Results: report total records analyzed and if data were excluded, or how you deal with missing data.                                                                                                      Authors’ Response: Thanks for this observation. This study is ecologic, implying the utilization of aggregate data.

4. Table 1 and 2: Include p value for Mortality Rate Ratio and the Total and the global calculation of each period. Preferably specify as a footnote, the meaning of E930-E936 , E870-E876  y Y40-Y84  (table 1), 812.1, 813.1, 814.1, 815.1, 816.1 , 811.1, 812.1, 813.1, 814.1, 815.1, 816.1 , V43.6  (Table 2).

Authors’ Response: The p value as random error quantification is not applicable in the CDC Wonder data.  However this study, as ecologic or aggregate data does not require this random error quantification as sampling variability and generalizability implying inference. These codes had been addressed in “notes and abbreviation section of the manuscript.

5. Figure 1 and 2: Review table format and scale in which they are presented. For example, the type of line presented in the graph is continuous and in the legend it is dotted.

Authors Response: Thanks. This had been addressed in the notes and abbreviation section, given the explanation of the line colors.

6. In Figure 2, it is not very clear which percentage points of change are described with those shown. You wrote: Notes and Abbreviations: The motor vehicular accidents prevalence ratio was slightly lower in the 282 first period (0.92, with blacks/AA, 8% less likely to experience this accident, but increased in the 283 second period (27% increased risk), and third period (17% increased risk). The percent change significantly lowered in the third period, 1999-2015. But in the figure, % change shows only 3 points  (0, 38.5 y -8.4)

Authors' Response: Thanks for this observation. We have addressed this kindly observe the section (Note and Abbreviations).

7.    Discussion: It is important to clearly discuss the quality of the information analyzed.                                                                                                      Authors’ Response: Thanks for this observation. However, it is not very clear with respect to the quality of information analyzed. However, we have utilized appropriate data from CDC in analysis and interpreting the data for the information utilized in this manuscript 

8.  Also discuss the reason for the change in percentage change (negative) of the last traffic accident period                                               

Authors’ Response: Thanks for this observation. However, due to the application of an ecologic data it is complex in the utilization of explanatory model in the process of understanding the outcome such as excess medial error and mistakes among black/AA children. In this direction the ability to incorporate confounding variable allows for the explanation of excess medical errors and mistakes among black/AA children. We have also observed based on the findings of studies the driving factors of medical errors and mistakes among black children.The data collection from different local department of health may reflect data unavailability at the giving point in time. This might explain the negativity.